# Comparison of Immediate Neuromodulatory Effects between Focal Vibratory and Electrical Sensory Stimulations after Stroke

**DOI:** 10.3390/bioengineering11030286

**Published:** 2024-03-17

**Authors:** Legeng Lin, Wanyi Qing, Yanhuan Huang, Fuqiang Ye, Wei Rong, Waiming Li, Jiao Jiao, Xiaoling Hu

**Affiliations:** 1Department of Biomedical Engineering, The Hong Kong Polytechnic University, Hong Kong, China; legeng.lin@connect.polyu.hk (L.L.); wanyi.qing@connect.polyu.hk (W.Q.); yanhuan731.huang@polyu.edu.hk (Y.H.); fuqiang.ye@connect.polyu.hk (F.Y.); rongwei@live.com (W.R.); deo-li@hotmail.com (W.L.); 2Research Institute for Smart Ageing (RISA), The Hong Kong Polytechnic University, Hong Kong, China; 3Department of Sport, Physical Education and Health, Hong Kong Baptist University, Hong Kong, China; jojojiao@hkbu.edu.hk; 4University Research Facility in Behavioral and Systems Neuroscience (UBSN), The Hong Kong Polytechnic University, Hong Kong, China; 5Joint Research Centre for Biosensing and Precision Theranostics, The Hong Kong Polytechnic University, Hong Kong, China; 6Research Centre on Data Science and Artificial Intelligence, The Hong Kong Polytechnic University, Hong Kong, China

**Keywords:** stroke, neuromodulation, focal vibratory stimulation (FVS), neuromuscular electrical stimulation (NMES), electroencephalography (EEG), cortical response, somatosensory impairment

## Abstract

Focal vibratory stimulation (FVS) and neuromuscular electrical stimulation (NMES) are promising technologies for sensory rehabilitation after stroke. However, the differences between these techniques in immediate neuromodulatory effects on the poststroke cortex are not yet fully understood. In this research, cortical responses in persons with chronic stroke (*n* = 15) and unimpaired controls (*n* = 15) were measured by whole-brain electroencephalography (EEG) when FVS and NMES at different intensities were applied transcutaneously to the forearm muscles. Both FVS and sensory-level NMES induced alpha and beta oscillations in the sensorimotor cortex after stroke, significantly exceeding baseline levels (*p* < 0.05). These oscillations exhibited bilateral sensory deficiency, early adaptation, and contralesional compensation compared to the control group. FVS resulted in a significantly faster P300 response (*p* < 0.05) and higher theta oscillation (*p* < 0.05) compared to NMES. The beta desynchronization over the contralesional frontal–parietal area remained during NMES (*p* > 0.05), but it was significantly weakened during FVS (*p* < 0.05) after stroke. The results indicated that both FVS and NMES effectively activated the sensorimotor cortex after stroke. However, FVS was particularly effective in eliciting transient involuntary attention, while NMES primarily fostered the cortical responses of the targeted muscles in the contralesional motor cortex.

## 1. Introduction

Over 50% of stroke survivors suffer from sensory impairments on the hemiplegic side [1], which consequently disrupt the intricate process of sensorimotor integration [2] and exacerbate the impairments of motor function [3]. Neuromuscular electrical stimulation (NMES) and focal vibratory stimulation (FVS) are the primary techniques used to deliver external somatosensory stimulation to specific muscles transcutaneously for sensorimotor rehabilitation after stroke [2,4,5]. NMES or FVS together with baseline motor rehabilitation has demonstrated efficacy for stroke rehabilitation [4,6,7]. This effectiveness could be attributed to the cortical process elicited through the integration of ascending sensory information from a targeted muscle to generate descending motor commands essential for motor initiation and planning, ultimately contributing to the enhancement of functional motor outcomes [2]. However, little is known about the differences between the transient neuromodulatory effects of these sensory stimulation techniques poststroke [8], which hinders their precise application in achieving effective neuroplasticity poststroke.

As an electrical stimulation to excitable cells, motor-level NMES is a technique in which electricity is used to evoke muscle contractions through the depolarization of motor nerves or muscle fibers [9]. It has been applied in routine poststroke interventions for enhancing muscular force, preventing muscle atrophy, and reducing muscle spasticity [10,11]. Sensory-level NMES with lower stimulation intensities than motor-level NMES mainly depolarizes the sensory neurons in the skin and muscles without eliciting muscle contraction [12]. It could improve muscular proprioception after stroke and reduce compensation from alternative muscular synergies [8,13]. Insausti-Delgado et al. reported that both motor- and sensory-level NMES applied to muscles could evoke event-related (de)synchronization (ERD/ERS) in the alpha and beta bands detected by electroencephalography (EEG) in unimpaired persons [14].

In comparison with NMES, FVS applied to a target muscle is more acceptable for stroke survivors because mainly the mechanoreceptors in the skin and muscles are activated during stimulation without wide recruitment of other sensory receptors (e.g., nociceptors, as in NMES) [12,15]. FVS can activate primary afferent endings (i.e., Ia afferents) in a muscle through mechanical deformation of the muscle spindles [16], which can increase muscular proprioception and suppress antagonist co-contraction [12]. FVS has also been found to ameliorate muscular spasticity during rehabilitation after stroke [7,17], with effects similar to those of manual massage.

Moreover, FVS evoked cortical activation comparable to that evoked by NMES in unimpaired individuals. For example, Hautasaari et al. reported that vibratory stimulation in healthy participants could evoke cortical activations similar to those evoked by electrical stimulation but with larger cortical areas being activated [12]. Event-related potentials (ERPs) evoked by FVS were adopted to investigate cognitive and somatosensory processes [18], e.g., P300, a positive wave with an onset ranging from 250 to 800 ms after a stimulation [19]. Bolton et al. reported that the P300 component evoked by vibratory stimulation was lower in amplitude with a longer latency in older adults than younger adults, mainly because of a slower cognitive response due to aging [20]. However, the understanding of the transient cortical responses of FVS applied to peripheral muscles is still lacking for stroke survivors, although preliminary rehabilitation effectiveness introduced by FVS has been reported in the literature [21,22]. In this study, we aimed to investigate and compare the immediate neuromodulatory effects of FVS and NMES at the cortical level in poststroke and unimpaired persons.

## 2. Materials and Methods

The transient cortical responses of participants with chronic stroke and the unimpaired controls were captured by EEG during FVS and NMES with different intensities to the forearm muscles. EEG was adopted in this study because it can reveal transient cortical responses to sensory stimulation with the advantages of high temporal resolution and cost-effectiveness compared to other neuroimaging techniques [23].

### 2.1. Participants

This study was approved by the Human Subjects Ethics Sub-Committee of the Hong Kong Polytechnic University before commencement (approval number: HSEARS20210320003). Participants after stroke were screened and recruited from local districts. The inclusion criteria were as follows: (1) at least 6 months after the onset of a unilateral lesion in the cortical or subcortical regions due to stroke [24]; (2) sufficient cognition to follow experimental instructions (Mini-Mental State Examination (MMSE) score > 23 [25]); (3) moderate-to-severe motor disability in the affected upper limb (15 < Fugl–Meyer assessment (FMA) < 45) [26,27]; (4) muscle spasticity scores ≤ 3 at the wrist and fingers, as measured by the modified Ashworth scale (MAS) [28]; (5) a normal-to-diminished protective sensation on the affected forearm under a threshold of 4.31 (2.0 g) as measured by the Semmes–Weinstein monofilament test [29,30] on the skin surface above the muscle union of the flexor carpi radialis (FCR) and flexor digitorum (FD) (i.e., FCR-FD) and the muscle union of the extensor carpi ulnaris (ECU) and extensor digitorum (ED) (i.e., ECU-ED) of the paretic forearm; (6) no neurological impairments except for stroke; and (7) right-handed before the stroke onset. The exclusion criteria for stroke participants were (1) poststroke pain, (2) epilepsy, (3) cerebral implantation, and (4) pacemaker implantation. Unimpaired participants were also recruited from the local districts; their inclusion criteria were right-handed and no history of neurological, psychiatric, cardiovascular, cognitive, or mental impairments. The clinical assessments mentioned above were carried out by an independent assessor who was blinded to the content of the study. Finally, 15 stroke participants (i.e., the stroke group) and 15 unimpaired participants (i.e., the control group) were recruited. All participants understood the study’s research information and signed the consent form before the experiment. Table 1 provides an overview of the demographic information of all the individuals involved in the study. The clinical scores of the stroke participants are presented in Table 2.

### 2.2. Experimental Setup

The experiment was carried out in a quiet environment where the temperature and humidity levels were regulated at 18–20 °C and 60 ± 5%, respectively. The experimental setup is shown in Figure 1A. The participant was requested to take a comfortable seated position facing a table, approximately 10 cm from the table edge to their torso. The participant was provided with an eye mask and a pair of earplugs to further minimize visual and auditory interference during the EEG recording. Then, they were instructed to reach their testing forearm forward, with the elbow resting at approximately 170° and supported by a cushion. The wrist and finger joints remained in a relaxed position. The contralateral upper limb was positioned naturally on the participant’s thigh. A 64-channel EEG cap was mounted on the scalp of the participant according to the standard 10-10 system (Figure A1A) [31]. The impedance between each EEG electrode and the scalp was maintained below 5 kΩ [32]. The EEG electrode cap was connected to amplifiers (BrainAmp MR, Brain Products Inc., Herrsching, Germany) and subsequently linked to a desktop computer for real-time EEG monitoring on one screen. A user interface was developed (LabVIEW 2015) and visualized on another screen to control the stimulation process (Figure 1A).

To enhance motor unit recruitment, the NMES electrode pairs (5 × 5 cm^2^, PALS Neurostimulation Electrodes, Axelgaard Manufacturing Co., Ltd., Fallbrook, CA, USA) were placed in the common area of the motor points for muscle bellies of the FCR-FD or ECU-ED muscle unions, given the close anatomical proximity of the two muscles in a muscle union (Figure A1B) [33,34]. Before electrode attachment, the skin was prepared to lower the skin–electrode impedance below 5 kΩ [32]. A miniature FVS vibration motor (model 310-122, Precision Microdrives, London, UK; 10 mm diameter, 3 mm height, 1.2 g weight; 11,500 rpm with 1.9 G amplitude at rated operating voltage 3 V) was gently pressed against the participant’s skin using medical tape between the cathode and anode of the NMES electrodes to ensure vibration transmission, with care taken to avoid sharp edges and the sensation of the electrical wiring [35]. We integrated the one-channel FVS into an NMES control box developed previously [27,33,36]. These NMES electrodes and the FVS vibration motor were controlled by the control box, with the outputs of one-channel FVS in a range of amplitudes from 0.7 G to 1.9 G below the threshold of the tonic vibration reflex without involuntary muscle contraction [37], and one-channel NMES generated alternating current in square pulses with a frequency of 40 Hz (i.e., 40 pulses per second), an amplitude of 70 V, and an adjustable pulse width ranging from 0 to 300 μs, allowing for different levels of stimulation intensity [36]. These FVS and NMES intensities were reported for sensory rehabilitation poststroke [14,38].

### 2.3. Experimental Protocol

Based on the above setup, we studied eight stimulating schemes, five FVS and three NMES schemes with different intensities. The FVS schemes had five intensities (Figure 1B): 1.9 G (FVS-1), 1.6 G (FVS-2), 1.3 G (FVS-3), 1.0 G (FVS-4), and 0.7 G (FVS-5); these intensities were reported to be effectively perceived and tolerable for the sustained stimulation of human participants [39]. The NMES schemes had three intensities (Figure 1C): (1) the motor threshold (NMES-1), identified as the initial twitching of the fingers [14]; (2) the representative sensory-level NMES (NMES-2), calculated as the median value between the perceptual and motor thresholds; and (3) the perceptual threshold (NMES-3), identified as the initial tingling sensation on the forearm [14]. Perceptual and motor thresholds were determined for each target muscle union by gradually increasing the NMES pulse width from 0 μs in steps of 1 μs. A duration of 10 s for sustained stimulation in all FVS and NMES schemes was selected as the stimulation block in this study (Figure 1D). Then, six stimulation blocks of the same scheme were arranged into a stimulation trial with a quiet period of at least 20 s between two consecutive blocks to minimize potential afferent adaptation [40]. During a trial, a participant was required to avoid active mental tasks, maintain a still posture, and minimize head and neck motions, e.g., ocular and swallowing motions. There were eight trials associated with the eight stimulation schemes applied to a muscle union in random order, forming the stimulation protocol (Figure 1D, Table 3). A 2 min interval between two consecutive trials was provided to a participant to move and rest. The stimulation protocol was applied sequentially to the four muscle unions of a participant in random order.

During a stimulation trial, the EEG signals were amplified with a gain of 10,000, digitized at an analog-to-digital sampling rate of 1000 Hz. For monitoring EEG in the experiment, the signals were notch-filtered from 49 Hz to 51 Hz and bandpass-filtered from 1 Hz to 100 Hz in the real-time processing. Meanwhile, the raw EEG signals in each stimulation trial were stored digitally after the sampling for later offline processing. The acquisition duration was 4 min for each trial, including the baseline and stimulation periods. The stimulation events were labeled by markers in the recorded EEG trials.

### 2.4. EEG Analysis

In the offline EEG analysis (Figure 2), temporal, spectral, and spatial features were analyzed to evaluate the cortical responses to different stimulation schemes after signal pre-processing. EEG processing and analysis were conducted with the EEGLAB (version 2022.0) [41] and Fieldtrip (version 20220603) [42] toolboxes with the latest update [43,44,45] using MATLAB R2019 (MathWorks, Natick, MA, USA).

#### 2.4.1. EEG Pre-Processing

In the offline processing, the recorded EEG signals in each stimulation trial were bandpass-filtered from 1 Hz to 100 Hz and notch-filtered from 49 Hz to 51 Hz digitally by a fourth-order Butterworth filter [46]. Independent component analysis (ICA) was applied to all EEG signals to minimize potential muscular artifacts [47,48]. EEG signals were further screened by visual inspection to remove artifacts. The electrode positions of the EEG data were flipped along the mid-sagittal plane for participants with left-hemisphere lesions so that the affected hemisphere was on the right side for all stroke participants [49]. The EEG signals of each trial were then segmented into six signal epochs with a duration of 15 s, corresponding to a 5 s baseline ahead of stimulation onset and a 10 s period during the stimulation block for later calculation [50] (Figure 2). Each epoch contained the EEG episodes from the 62 channels. There were 178,560 EEG episode samples in the respective stroke and control groups (15 participants × 2 arms × 2 muscle unions × 8 schemes × 6 epochs × 62 episodes).

#### 2.4.2. EEG Temporal, Spectral, and Spatial Features

Four EEG features were used to investigate the cortical response: the ERP in the temporal domain, the relative spectral power (RSP) in the spectral domain, the event-related spectrum perturbation (ERSP) in the time–frequency domain, and the ERD/ERS topography in the spatial domain at the cortical level (Figure 2). These EEG features were calculated for each episode in an epoch. The ERP waveform of an episode was obtained from the baseline correction using Equation (1):(1)$$ERP\left(t\right)=p_{block}\left(t\right)-\bar{p_{baseline}\left(t\right)}$$
where $p_{block}\left(t\right)$ is the EEG waveform during the stimulation block and $\bar{p_{baseline}\left(t\right)}$ is the mean value over the baseline period. The ERPs on the Fz, Cz, and Pz electrodes were averaged to obtain P300 [19]. To enable subsequent statistical analysis, the peak amplitude relative to the baseline and the peak latency after stimulation onset were computed for each participant’s P300. The period of interest for RSP analysis was defined as the time window during which P300s were significantly different. The RSP of an episode was calculated by Equation (2):(2)$$RSP_{band}=\frac{\int_{F_{1}}^{F_{2}}p_{block}\left(f\right)df}{\int_{1}^{100}p_{block}\left(f\right)df}-\frac{\int_{F_{1}}^{F_{2}}p_{baseline}\left(f\right)df}{\int_{1}^{100}p_{baseline}\left(f\right)df}$$
where $p_{block}\left(f\right)$ is the power spectral density during the period of interest in the stimulation block; $p_{baseline}\left(f\right)$ is the power spectral density during the baseline; and *F*_1_ and *F*_2_ are the cutoff frequencies of the EEG frequency bands, which are the theta (θ, 4~8 Hz), alpha (α, 8~12 Hz), beta (β, 13~30 Hz), and gamma (γ, 30~100 Hz) bands in this study [23]. Then, the RSPs were averaged across the episodes from the whole-brain channels (i.e., whole-brain RSP) as practiced previously [23] and across the episodes from channels on the sensorimotor cortex contralateral to the stimulated side (i.e., contralateral sensorimotor RSP) to quantify the response in the contralateral sensorimotor areas. The RSP was narrowed to the predefined period of interest and averaged on the frequency bands for quantification, while the ERSP was used to analyze the 2-dimensional cortical response with respect to continuous time and frequency. The ERSP of an episode for each EEG channel was obtained from baseline normalization using Equation (3):(3)$$ERSP\left(f,\;t\right)=\frac{S_{block}\left(f,\;t\right)-{µ}_{baseline}\left(f\right)}{\sigma_{baseline}\left(f\right)}$$
where $S_{block}\left(f,\;t\right)$ is the time–frequency spectrum during the stimulation block, and ${µ}_{baseline}\left(f\right)$ and $\sigma_{baseline}\left(f\right)$ are the mean and standard deviation, respectively, of the baseline spectrum. A baseline permutation statistical method (2000 times) with false discovery rate correction for multiple comparisons was adopted, with a significance level of 0.05 [51]. ERSPs of C3 and C4 were used as representative channels from the bilateral hemispheres [14,52] to investigate lateralization of cortical activation during sensory stimulation to both arms. ERD/ERS topographies were used to evaluate the cortical distribution patterns of the peak values in each EEG band after the stimulations for the two subject groups. Based on the ERSP, the ERD/ERS of each channel was calculated by Equation (4):(4)$$ERD/ERS=\frac{1}{K}\underset{f\in F}{\sum }\underset{t\in T}{\sum }EPSP\left(f,t\right)$$
where *F* is the frequency band (i.e., theta, alpha, beta, or gamma bands), *T* is the analyzed latency within the stimulation block, and *K* is the total number of time–frequency bins within the time–frequency window [51]. The FVS-1 and NMES-2 stimulation schemes were chosen as representative stimulations in the analyses of ERSP and ERD/ERS topography as these two intensities evoked the strongest cortical ERPs in the sensory stimulations.

For each participant, an EEG feature was averaged across the six repeated blocks in a stimulation trial. This operation resulted in an averaged EEG feature with respect to a stimulation scheme on a muscle union of a participant, which was used as an experimental reading unit for statistical analysis.

### 2.5. Statistical Analysis

The statistical analyses (Figure 3) were conducted for the monofilament scores, perceptual/motor NMES thresholds, ERPs, RSPs, and ERD/ERS topography. The features were first compared with respect to two independent factors: (1) the stimulation scheme (FVS-1,2,3,4,5 and NMES-1,2,3) and (2) the target muscle union (ECU-ED and FCR-FD muscle unions). The features were further combined and compared based on three factors: (1) the type of stimulation (FVS or NMES) used to determine the differences between the vibratory and electrical stimulation types; (2) the target arm (dominant or nondominant arm) used to determine the difference between the two sides; and (3) the group (control or stroke group) used to determine the changes in the cortical response after stroke. The corresponding abbreviations of the five factors are listed in Table 3.

The Shapiro–Wilk normality test with Lilliefors correction was first performed. The amplitude and latency of the P300 peak and the RSP were normally distributed (*p* > 0.05). Nonparametric tests were adopted for non-normally distributed data; Mann–Whitney U tests were conducted on the monofilament scores and perceptual/motor NMES thresholds for intra-group comparison between the two target muscle unions and two target arms, and Wilcoxon signed-rank tests were conducted for inter-group comparison based on the target muscle union. In addition, the ERP amplitudes during the stimulation blocks were compared using cluster-based permutation tests (2000 permutations) for intra-group comparisons among the eight stimulation schemes and between the two muscle unions, as well as inter-group comparisons among the four target arms. Paired and independent *t*-tests were applied to the respective intra-group and inter-group comparisons of the P300 peak’s amplitude and latency between the FVS and NMES stimulation types and between the stroke and control groups. Moreover, the ERD/ERS topographies during representative FVS and NMES were compared between the stroke and control groups by cluster-based permutation tests (2000 permutations) to investigate poststroke spatial alterations during FVS and NMES.

Parametric tests were adopted for RSP in each frequency band. Intra-group comparisons of the RSP were conducted for each group. Two-way mixed analyses of variance (ANOVAs) were used to evaluate the differences in RSP with respect to the factors of the stimulation scheme and target muscle union, as well as the stimulation scheme and target arm. Then, a one-way ANOVA with repeated measures (RM) was used to compare the RSP among the eight stimulation schemes with the Bonferroni post hoc test. Paired *t*-tests were used to compare the RSP between different target muscle unions, target arms, or stimulation types.

Next, inter-group comparisons of the RSP were conducted between groups. A two-way mixed ANOVA was conducted to evaluate the differences in RSP with respect to the factors of the stimulation scheme and group. Independent *t*-tests were subsequently conducted to compare the RSP between the stroke and control groups based on the stimulation scheme, target muscle union, target arm, and stimulation type. The cluster-based permutation test was executed using the FieldTrip toolbox with the Monte Carlo method [44]. Other statistical analyses were performed utilizing SPSS 24.0 (2016). The statistical significance level was set at 0.05 in this work, with levels of 0.01 and 0.001 also indicated.

## 3. Results

### 3.1. Monofilament Test and NMES Thresholds

After the subject screening, 15 out of 17 stroke survivors (more than 88%) satisfied the criteria for a normal-to-diminished protective sensation (<4.31) in the monofilament test. Figure 4 shows the monofilament scores and perceptual/motor NMES thresholds of all muscle unions in the stroke and control groups. The detailed mean values and standard errors (SEs) are summarized in Table A1, Table A2 and Table A3, along with the Mann–Whitney U test or Wilcoxon signed-rank test probabilities and the estimated effect sizes (EFs) [53]. The only significant finding in the monofilament score (Figure 4A) was that the sensitivity of the unaffected ECU-ED (UE) of the stroke group was significantly lower than that of the right ECU-ED (RE) of the control group (*p* = 0.009). No significant differences were observed in the intra-group comparisons between the target arms and target muscle unions within the stroke and control groups (*p* > 0.05).

According to the results of the inter-group comparison of NMES thresholds (Figure 4B,C), the perceptual thresholds of the affected ECU-ED (AE) and FCR-FD (AF) muscle unions in the stroke group were significantly higher than those of the left ECU-ED (LE) and FCR-FD (LF) muscle unions in the control group (*p* = 0.009 and 0.037). The motor NMES threshold of AF muscle union in the stroke group was significantly higher than that of the LF in the control group (*p* = 0.016). According to the intra-group comparisons, the perceptual and motor NMES thresholds of the RE muscle union were significantly higher than those of the RF muscle union in the control group (*p* = 0.010 and 0.004). The motor NMES thresholds of the UE muscle union were significantly higher than those of the unaffected FCR-FD (UF) muscle union in the stroke group (*p* = 0.002). Moreover, the perceptual and motor NMES thresholds of the affected arm (including AE and AF muscle unions) were significantly higher than those of the unaffected arm (including UE and UF muscle unions) in the stroke group (*p* < 0.001). In contrast, no significant difference was observed between the left and right arms in the control group (*p* = 0.160 and 0.246).

### 3.2. P300 in the ERP Response to FVS and NMES

Figure 5A,B display the P300 waveforms averaged across subjects with respect to the factors of the group, stimulation scheme, and target arm. No intra-group significance was found between the two target muscle unions on the same arm (*p* > 0.05). Figure 5C illustrates the comparison between the peak amplitude and latency of the P300 waves. The corresponding values for the amplitude and latency can be found in Table A4, along with the probability and EFs for both the paired and independent *t*-tests. According to the intra-group comparison of the eight stimulation schemes (Figure 5A), significant P300 differences in amplitude were found from 339 to 706 ms in the control group and from 375 to 688 ms in the stroke group (*p* < 0.05). P300 responses to FVS were activated earlier with a higher peak amplitude in the control group than NMES (*p* < 0.001, Figure 5C) and were activated earlier (*p* < 0.001) without a significant difference in peak amplitude in the stroke group than NMES. There was also a “stimulation-intensity-dependent response” tendency in which a higher intensity evoked a quicker response and a higher amplitude of P300.

According to the inter-group comparison among the target arms of the stroke and control groups based on the stimulation scheme (Figure 5B), significant differences were observed from 363 to 478, 385 to 459, 389 to 467, 379 to 528, and 451 to 530 ms for responses to the FVS-1, 2, 3, 4, and 5 schemes, respectively (*p* < 0.05). The P300 peak amplitude in the control group during FVS was significantly higher than that in the stroke group (*p* < 0.001, Figure 5C). In contrast, no significant difference between the stroke and control groups was observed in all the NMES schemes (*p* > 0.05). Overall, the significant period of P300 was between 300 and 750 ms, which was considered the period of interest in the RSP analysis.

### 3.3. RSP Response on the Contralateral Sensorimotor Cortex

No significant differences among stimulation schemes were observed in the whole-brain RSP (*p* > 0.05). The RSP results reported here (Figure 6 and Figure A2) depict the contralateral sensorimotor RSP averaged across subjects during the period of interest. Table A5, Table A6 and Table A7 list the probability and EFs for the two-way mixed ANOVAs. No significant interaction was found in any of the frequency bands, and no intra-group significant difference was found between the target muscle unions (*p* > 0.05). Table A8, Table A9 and Table A10 provide the detailed means and SEs of RSP with respect to the stimulation scheme, target muscle union, stimulation type, target arm, and group, in addition to the probabilities and EFs of the one-way ANOVA with repeated measures, as well as independent and paired *t*-tests.

Figure 6A shows the RSP comparison with respect to the stimulation scheme, target arm, and group. Intra-group significant differences in the beta and gamma bands were observed among stimulation schemes in the right arm of the control group (*p* < 0.001). According to the post hoc test, the NMES-3 scheme yielded significantly lower RSPs than the FVS-1, 2, 3, and 4 schemes in the beta band (*p* < 0.05, adjusted by Bonferroni correction). In the gamma band, the NMES-3 scheme yielded significantly higher RSPs than the FVS-1, 2, 3, 4, and 5 schemes, and the NMES-2 scheme yielded significantly higher RSP than the FVS-2 and 4 schemes (*p* < 0.05, adjusted by Bonferroni correction). No intra-group significant differences were found between the stimulation schemes for the control group in the theta or alpha band or for the stroke group in any band (*p* > 0.05). According to the results of the inter-group comparison, the overall RSP in the stroke group was significantly lower than that in the control group in the alpha and beta bands (*p* < 0.001 and =0.012). In particular, the RSPs of the stroke group were significantly higher than those of the control group in the nondominant arms (i.e., A and L) during all stimulation schemes and in the dominant arms (i.e., U and R) during the FVS-1, 2, 4, 5, and NMES-1, 2 schemes in the alpha band (*p* < 0.05), as well as in the nondominant arms during the FVS-5 and NMES-2, 3 schemes and in the dominant arms during the NMES-3 scheme in the beta band (*p* < 0.05). No inter-group significant differences based on the stimulation scheme or target arm were observed in the theta or gamma band (*p* > 0.05).

Figure 6B shows the RSP comparison with respect to the target arm and group. According to the intra-group comparison, the RSP in the theta band was significantly higher in the nondominant arm than in the dominant arm in the control group (*p* = 0.004). No intra-group significant differences between the target arms were observed in the alpha, beta, or gamma band (*p* > 0.05). According to the results of the inter-group comparison, the RSP in the theta band was significantly lower in the nondominant arm in the stroke group than in the control group (*p* = 0.009). In the alpha and beta bands, both arms of the stroke group had significantly higher RSPs than those of the control group (*p* < 0.05). No inter-group significant differences based on the target arm were observed in the gamma band (*p* > 0.05).

Figure 6C shows the RSP comparison with respect to the stimulation type and group. According to the intra-group comparison, the RSP of the control group during FVS was significantly higher than that during NMES in the theta and beta bands (*p* < 0.001 and =0.023) but was significantly lower than that during NMES in the gamma band (*p* < 0.001). No intra-group significant differences between FVS and NMES were observed for the control group in the alpha band or for the stroke group in any of the bands (*p* > 0.05). According to the results of the inter-group comparison, the RSP during FVS in the theta band was significantly lower in the stroke group than in the control group (*p* = 0.005). In the alpha and beta bands, the RSP of the stroke group during FVS and NMES was significantly higher than that of the control group (*p* < 0.01). No inter-group significant differences were observed in the gamma band (*p* > 0.05).

### 3.4. ERSP Response on the Bilateral Sensorimotor Cortex

Figure 7 shows the ERSP at C3/C4 averaged across subjects with respect to the stimulation scheme, target arm, and group. In the control group, compared with the resting baseline, FVS and NMES evoked significant ERD in the alpha and beta bands (i.e., αERD and βERD) and significant ERS in the theta band (i.e., θERS) after stimulation onset (*p* < 0.05). For both FVS and NMES, compared to those in the control group, the stroke group exhibited decreased amplitudes and restricted distributions in ERD/ERS across the frequency bands. In addition, bilateral ERD/ERS could be observed in the control group on both arms. However, when the affected arm of the stroke group was stimulated, the ERD/ERS was more intensive in the ipsilateral/contralesional hemisphere (Figure 7C) than in the contralateral/ipsilesional hemisphere (Figure 7A).

### 3.5. Spatial Distribution of ERD/ERS Topography

Figure 8 displays the averaged ERD/ERS topographies and comparisons between the stroke and control groups in response to FVS-1 and NMES-2. In Figure 8A, for both FVS and NMES, the stroke group showed lower holistic ERD/ERS in restricted cortical areas than the control group. The θERS mainly occurred on the bilateral central area in the control group, and additional recruitment over the ipsilateral parietal area could be activated during FVS compared to NMES. No θERS was observed in the stroke group. The αERD occurred mainly on the bilateral central area in the control group, but on the contralesional hemisphere in the stroke group. The αERD peaks in the stroke group shifted from the central area to the contralesional parietal–occipital area compared with those in the control group. The βERD occurred mainly on the bilateral central area in both subject groups. As shown in Figure 8B, in comparison with that in the control group, the θERS in the stroke group was significantly suppressed on the bilateral central–parietal–occipital area in response to FVS (*p* = 0.003) and mainly on the paramedian central area in response to NMES (*p* = 0.005). The αERD was significantly weakened mainly on the ipsilesional central–parietal area during both FVS and NMES (*p* = 0.003). The βERD was significantly diminished on the paramedian central–frontal area during FVS (*p* = 0.016) and on the ipsilesional central–frontal area during NMES (*p* = 0.018).

## 4. Discussion

### 4.1. Altered Perceptual Sensitivity after Stroke

The monofilament test mainly assesses the tactile function of the skin. The increased perceptual sensitivity in the unaffected ED-ECU of stroke survivors revealed by the monofilament results (Figure 4A) could be related to the sensory stimulation resulting from the habitually preferred use of the unaffected upper limb in chronic stroke. The lack of discrimination between the limbs and between the subject groups except UE vs. RE (Figure 4A) suggested that skin tactile function was basically preserved in the poststroke participants recruited in this work. In contrast to the preserved tactile function of the skin, combined sensorimotor impairment poststroke was revealed by the NMES evaluations. Both perceptual and motor thresholds in the affected arm were higher than those in the other arms, indicating weakened afferent and efferent functions in the target muscles after stroke (Figure 4B,C). The between-group differences in sensory and motor NMES shown in Figure 4B,C revealed combined effects of the skin and muscular functions on the impaired perceptual sensitivity after stroke, while the differences in the skin tactile function were not significant. It implied that the sensory impairments after stroke in the targeted positions in this study could be mainly related to the sensory deficiency in muscles.

### 4.2. Earlier P300 Evoked by FVS Than NMES

For both subject groups, the P300 exhibited a stimulation-intensity-dependent trend at various intensities of FVS and NMES (Figure 5A). A direct quantitative association was demonstrated between brain activation and stimulation intensity, which was consistent with previous reports on the response to sensory stimulation in unimpaired persons measured by EEG [14], magnetoencephalography [54], functional magnetic resonance imaging [55,56], and transcranial magnetic stimulation [57]. Compared to NMES, the FVS in the control group evoked significantly higher peak amplitudes with shorter latencies in the P300 response (Figure 5A,C). This could be related to the fact that FVS can activate selective mechanoreceptors (mainly Ia afferent endings in the muscle spindles and Pacini receptors in the skin) with synchronous action potential in the afferent pathway [16], while NMES can bypass mechanotransduction and directly elicit wider recruitment of diverse sensory receptors and nerve fibers, generating varying conduction velocities with receptor delays in the afferent pathway compared to FVS [58,59]. In addition, previous studies on sensory stimulation have suggested that a shorter latency in ERP is associated with less effort/attention to perform the sensation task [60], and a higher peak amplitude indicates more favorable skin interactions with fewer distractions to achieve better attention [18]. It indicated that FVS was a more favorable sensory stimulus than NMES in this study, requiring less effort of attention and fewer cognitive resources for perception. During FVS, P300 peaks were lower in the stroke group than in the control group (Figure 5B,C). This could be related to an impaired afferent pathway and reduced neural resources at the cortical level after stroke [19,20]. However, FVS still demonstrated a faster response in evoking P300 than NMES in the stroke group (Figure 5A,C), mainly because of the homogeneous recruitment of mechanoreceptors in the skin and muscle spindles.

### 4.3. Spectral Features of Cortical Responses to FVS and NMES

The power suppression (e.g., in RSP) or desynchronization (i.e., ERD) of alpha and beta rhythms over the sensorimotor cortex has been extensively employed as a characteristic of neural activation during somatosensory tasks in unimpaired participants [8,14,61]. For example, NMES above the motor threshold has been reported to induce significant ERD in the alpha and beta bands over the sensorimotor cortex compared with the resting baseline [14,62]. In addition, by depolarizing peripheral sensory neurons in the skin and muscles without causing muscular contraction, sensory-level NMES also effectively conveys proprioception and induces significant ERD in the alpha and beta bands across sensorimotor areas through the afferent pathway [8]. In this study, the weakened activation level of the sensorimotor cortex in the alpha and beta bands (Figure 7) suggested stroke-induced sensory deficiency through the afferent pathway to the sensorimotor cortex. However, the ERD in the related bands in response to sensory-level NMES and FVS (1.9 G) still could be significantly differentiated from the baseline in the stroke group, with similar patterns between the two different stimulation types (Figure 7, stroke group). Similar to the ERD patterns, the alpha- and beta-RSPs over the contralateral sensorimotor cortex evoked by the FVS intensities had equivalent values to those by NMES with the intensities above the perceptual threshold (Figure 6). These results suggested that FVSs could effectively evoke neural responses in the lesioned sensorimotor cortex through the afferent pathway in stroke survivors as NMESs above the perceptual threshold, which was considered the rehabilitative potential to induce neuroplastic modifications within the sensorimotor cortex poststroke [8,62].

Compared to those in the control group, the holistic lowered alpha- and beta-RSP for both affected and unaffected arms of stroke survivors (Figure 6B) indicated bilaterally weakened sensorimotor activation after stroke. As indicated in the study by Genna et al., the cortical processing of sensory stimulation involves the bilateral hemispheres during tactile stimuli to unilateral arms, i.e., the sensory responses are fused through inter-hemispheric pathways in unimpaired persons [40]. Similarly, in this study, the bilaterally weakened sensorimotor activation level in the stroke group also indicated the impact of impaired inter-hemisphere neural networks on the sensation of both affected and unaffected limbs, i.e., bilateral sensory deficiency, due to lesions in the ipsilesional hemisphere.

Moreover, the significant αERD response induced by prolonged FVS and NMES, i.e., 10s, in this study was mainly concentrated within the first 4 s after the stimulation onset in the unimpaired controls, but the αERD response was nonsignificant thereafter (Figure 7). It was related to the sensory adaptation to a sustained stimulus with attenuated cortical responses [14]. In contrast to that in the control group, the shorter duration of αERD in the stroke group suggested an earlier adaptation to external somatosensory stimuli poststroke because of a rapid neuronal desensitization in the afferent pathway [63]. The neuronal desensitization in the sensory pathway covered more than the mechanotransduction stimulated by FVS, as the αERD evoked by NMES of the stroke group also demonstrated similar shortened durations.

Significant θERS was observed in the control group upon the onset of both FVS and NMES (Figure 7). However, this response was weak in the stroke group. Theta-band oscillations could reflect changes in attention given to new information/stimulation [64]. Several studies have reported that power enhancement or synchronization in theta rhythms within 500 ms after the onset of an external stimulus mainly indicates transient involuntary attention given to this new sensory stimulus [40,65]. In the control group, the significantly higher theta RSP during FVS than that during NMES (Figure 6C) indicated that FVS was more effective at evoking involuntary attention than NMES. Similar findings of the distinctions between FVS and NMES in theta oscillations could be observed for the stroke group (Figure 6C and Figure 7). However, they were not statistically significant, primarily due to the diminished temporary involuntary attention allocated to these sensory inputs after stroke.

It was also observed that NMES could arouse higher gamma RSP than FVS in the control group (Figure 6C). This difference might be related to the extent of unpleasant perceptions associated with the different stimulation types [66]. Even at the sensory levels, NMES could cause burning or tingling sensations, while the intensities of FVS in this work mainly achieved sensory feelings similar to those of manual stimulations, e.g., skin tapping or light pressing. The unpleasant perception during NMES was related to a wide recruitment of diverse sensory receptors and nerve fibers, including nociceptors, which accounted for the significantly higher gamma RSP than that of FVS eliciting homogeneous mechanoreceptors in the skin and muscle spindles [12].

### 4.4. Topographical Patterns of Cortical Responses to FVS and NMES

Transient cortical modulation in the sensorimotor area has been found in unimpaired persons during transcutaneous electrical sensory stimulation to muscles [12,14]. The transmission of the stimulation inputs through the afferent pathway, from peripheral sensory neurons in the muscles and skin to the spinal cord, can activate the primary motor cortex (M1) via Brodmann’s area 3a in the primary somatosensory cortex (S1) [62]. In this study, compared to the sensory-level NMES, FVS (1.9 G) activated similar sensorimotor areas in the alpha and beta bands in both subject groups (Figure 8A). These findings suggested that FVS could achieve cortical recruitment in the sensorimotor cortex via the afferent pathway similar to that of NMES, with similar intensities of the RSPs in the alpha and beta bands (Figure 6).

However, the cortical areas recruited by FVS in the θERS topography were larger than those recruited by NMES in the control group, with additional recruitment covering the ipsilateral parietal area (Figure 8A). In a study of cortical somatosensory responses in unimpaired persons by Hautasaari et al., it was reported that brain activation on the sensorimotor cortex could be elicited more widely by mechanical stimulation than by electrical stimulation [12]. This is mainly because mechanical stimulation elicits more homogeneous mechanoreceptors in the skin and muscle spindles and additionally activates the ipsilateral somatosensory cortex, drawing more involuntary attention compared to electrical stimulation. In this study, the results of wider recruitment in the θERS topography together with the higher theta RSP (Figure 6C) obtained in the control group over the sensorimotor cortex suggested that FVS was a more effective stimulation for recruiting cortical resources for transient involuntary attention than NMES. The additional recruitment over the ipsilateral parietal area suggested that the posterior association area was recruited for somatosensory perception by FVS, which indicated better spatial attention/awareness of the body elicited by FVS than by NMES. However, θERS in the cortex of the stroke group could not be significantly related to poststroke numbness, mainly because of the weakened afferent signals evoked by both stimulation types compared to the control group (Figure 6C, Figure 7, and Figure 8B).

The stroke group exhibited contralesional compensation and domination in the cortical responses compared to those of the control group. Consistent with previous findings in unimpaired individuals [40], bilateral activation of the sensorimotor cortex was observed in the control group during FVS and NMES (Figure 7 and Figure 8A, control group) because the sensory responses are normally fused through inter-hemispheric pathways. In contrast, the αERD over the contralesional hemisphere remained after stroke (Figure 7C, stroke group, and Figure 8A), but the αERD over the ipsilesional hemisphere was significantly weakened (Figure 8B). This observation agreed with previous findings that the reorganization of somatosensory neurocircuits after stroke involves cortical recruitment concentrated in contralesional cortical regions, i.e., contralesional compensation [67,68]. Furthermore, αERD peaks shifted to the contralesional parietal–occipital area in the stroke group (Figure 8A), which revealed that the contralesional somatosensory association cortex and visual cortex outside the S1 were involved in the sensory process poststroke [23,69]. This implied that stroke survivors exerted additional effort in the somatosensory association cortex and engaged supramodal neural networks in the visual cortex related to spatial attention, as part of their contralesional compensation mechanism. This allowed them to process and analyze sensory information from S1 in response to sensory stimuli after stroke.

Significantly attenuated βERD responses were observed in the stroke group over the frontal–parietal area, i.e., motor cortex including M1, supplementary motor area (SMA), and premotor cortex (PMC) (Figure 8B). Interestingly, asymmetric cortical responses were captured in response to NMES in contrast to FVS. Beta oscillations play a crucial role in the closed-loop neural network for the transmission of motor-related information from the M1 to the muscles and back to the M1 via somatosensory pathways in corticospinal communications [70]. Corbet et al. indicated that the beta oscillations enhanced by electrical sensory stimulation could be interpreted as stimulation-fostered muscle representation in the motor cortex according to the beta closed-loop neural network [8]. In this study, for the ipsilesional hemisphere, beta oscillations were significantly attenuated in response to both FVS and NMES after stroke (Figure 8B), indicating impaired beta closed-loop neural network wiring the ipsilesional motor cortex after stroke in response to the two sensory stimulation types. However, in the contralesional hemisphere, in contrast to FVS eliciting significantly weakened beta oscillations after stroke, NMES could evoke comparable beta oscillations to those in the control group (Figure 8B). This finding suggested that the sensory-level NMES fostered cortical responses of the targeted muscles by recruiting additional contralesional pathways for motor enhancement in comparison with that of FVS. This observation might also indicate that NMES could be more prone to trigger contralesional compensation in motor restoration than FVS.

### 4.5. Limitations and Future Work

One potential limitation of this study was the relatively small sample size. The recruitment of participants continued until significant differences in the key EEG parameters were observed between FVS and NMES among the groups. Fifteen participants were finally recruited in each group, and the statistical significances achieved in the study showed sufficient effect sizes to reach conclusions. Another limitation of this study was the age disparity between the stroke and control groups. Despite the control group having a higher mean age than the stroke group, the significant inter-group differences in the EEG patterns remain evident. The significant findings suggested that neurological impairments introduced by stroke were the dominant factors over aging [71] on the cortical responses to these sensory stimuli.

In addition to focal stimulation on a single target muscle investigated in the study, the application of distributed electrical and vibratory stimulations to multiple muscles has also demonstrated rehabilitative effects in patients following neurological disorders, e.g., whole-body vibration [72] and the Electrosuit [73]. Future studies will be conducted on developing selective stimulation techniques with distributed vibratory stimulation and investigating the related neuromodulatory effects on muscle groups (e.g., agonist and antagonist, proximal and distal) with larger sample sizes of participants.

## 5. Conclusions

In this study, the immediate neuromodulatory effects of FVS and NMES were compared by EEG measurement on the cortical responses in individuals with chronic stroke and unimpaired controls. The results in alpha and beta oscillations of the stroke group revealed that both FVS at 1.9 G and NMES above the perceptual threshold effectively activated the sensorimotor cortex and demonstrated similar patterns in stroke-induced effects, characterized by bilateral sensory deficiency on both affected and unaffected sides, early adaptation to external somatosensory stimuli, and contralesional compensation additionally eliciting the parietal–occipital cortex. However, FVS was found to be particularly effective in eliciting transient involuntary attention, as evidenced by faster response in P300 and higher theta oscillations, mainly because of the homogeneous recruitment of mechanoreceptors in the afferent pathway by FVS. In contrast, sensory-level NMES primarily fostered cortical responses of the targeted muscles in the motor cortex by recruiting additional contralesional pathways for motor enhancement, as observed through beta oscillations. Overall, by elucidating the specific cortical responses involved in FVS and NMES, this study contributes to our understanding of their potential applications in stroke rehabilitation and provides valuable insights for developing more tailored interventions in the future.

## Figures and Tables

**Figure 1 bioengineering-11-00286-f001:**
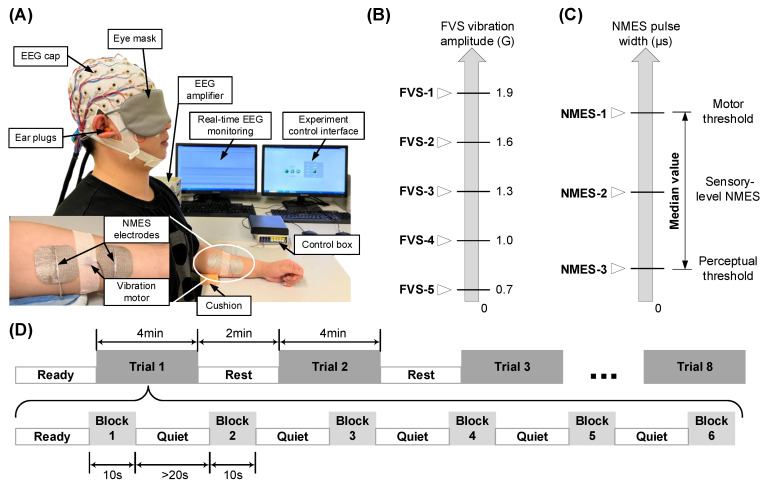
(**A**) Experiment setup. (**B**) Identification of 5 intensities in FVS schemes. (**C**) Identification of 3 intensities in NMES schemes according to the perceptual and motor threshold. (**D**) Experiment protocol timing for a target muscle. The 8 different schemes of FVS and NMES were randomly delivered in 8 trials. The same protocol was conducted on the both FCR-FD and ECU-ED muscle unions of the right and left arms of each participant.

**Figure 2 bioengineering-11-00286-f002:**
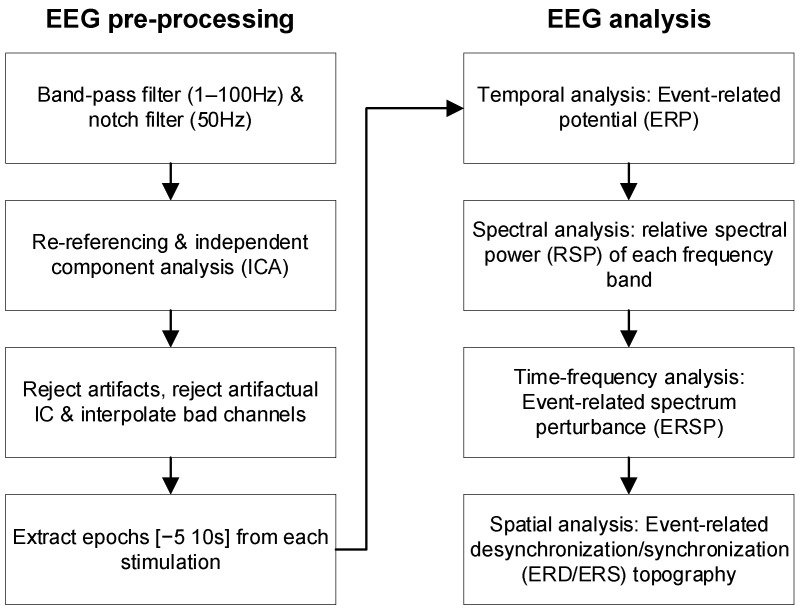
Flow chart of EEG pre-processing and analysis.

**Figure 3 bioengineering-11-00286-f003:**
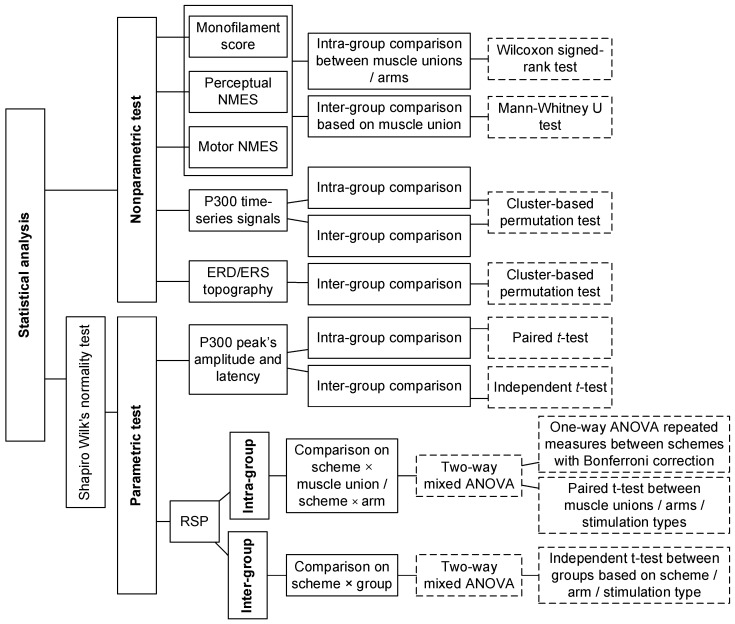
The logic flow of statistical analysis.

**Figure 4 bioengineering-11-00286-f004:**
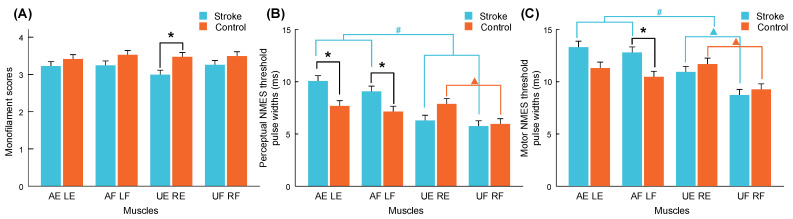
Monofilament scores (**A**) and pulse widths of perceptual (**B**) and motor (**C**) NMES thresholds on forearm muscle unions of the stroke and control groups, presented as the mean with SE (error bar). Significant inter-group differences based on muscle unions are indicated by “*” (*p* < 0.05, Mann–Whitney U test). Significant intra-group differences between ECU-ED and FCR-FD muscle unions and between dominant and nondominant arms are indicated by “▲” and “#” (*p* < 0.05, Wilcoxon signed-rank test). (AE, affected ECU-ED; AF, affected FCR-FD; UE, unaffected ECU-ED; UF, unaffected FCR-FD for stroke group. LE, left ECU-ED; LF, left FCR-FD; RE, right ECU-ED; RF, right FCR-FD for control group).

**Figure 5 bioengineering-11-00286-f005:**
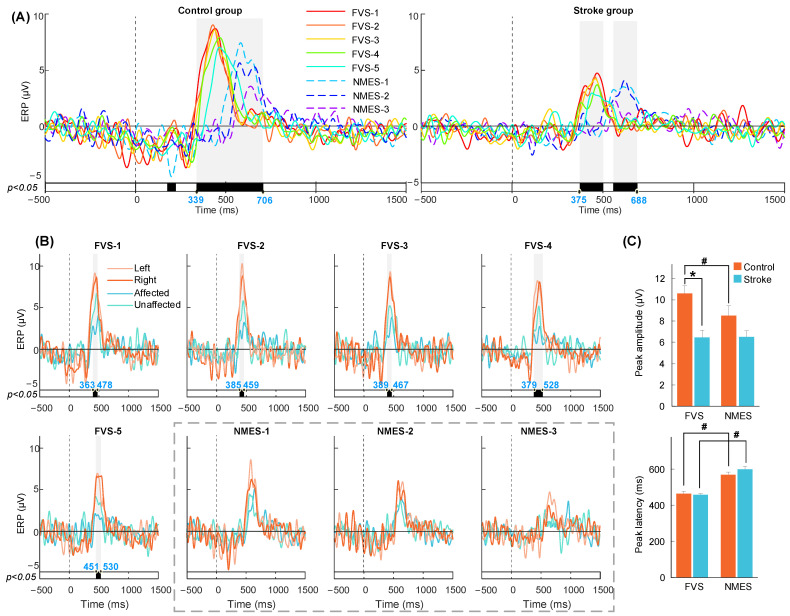
P300 in ERP averaged across subjects on the Fz, Cz, and Pz electrodes. The vertical dashed lines at 0 s mean the onset of stimulations. (**A**) Intra-group comparisons among different schemes. (**B**) Inter-group comparisons among target arms of stroke and control groups based on stimulation scheme. The bold line means the significantly different periods between the amplitude of ERPs compared within a sliding time window (*p* < 0.05, cluster-based permutation test). The shading means P300 with statistical significance, of which the start and end time points are shown in blue. (**C**) Comparison of P300 peak amplitude and latency. Significant inter-group and intra-group differences are indicated by “*” and “#” (*p* < 0.05, independent *t*-test and paired *t*-test).

**Figure 6 bioengineering-11-00286-f006:**
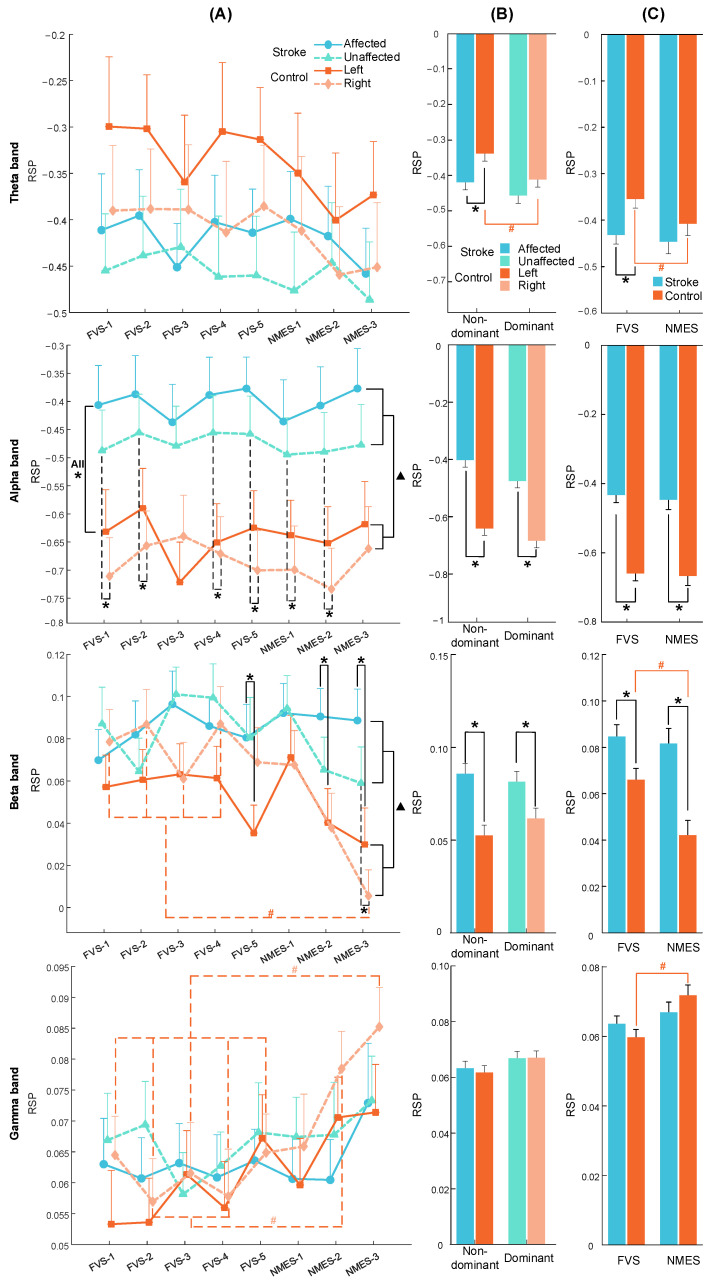
RSP averaged across subjects on the contralateral sensorimotor cortex in the theta, alpha, beta, and gamma bands during the period of interest. The RSP values are presented as the mean with SE (error bar). RSP comparison with respect to (**A**) the stimulation scheme, target arm, and group. Significant intra-group differences between schemes are indicated by “#” (*p* < 0.05, one-way ANOVA repeated measures with Bonferroni post hoc tests). Overall significant inter-group differences are indicated by “▲” (*p* < 0.05, two-way mixed ANOVA). Significant inter-group differences based on target arm are indicated by “*” (*p* < 0.05, independent *t*-test). RSP comparison with respect to (**B**) the target arm and group and (**C**) the stimulation type and group. Significant intra- and inter-group differences are indicated by “#” and “*” (*p* < 0.05, paired and independent *t*-test), respectively.

**Figure 7 bioengineering-11-00286-f007:**
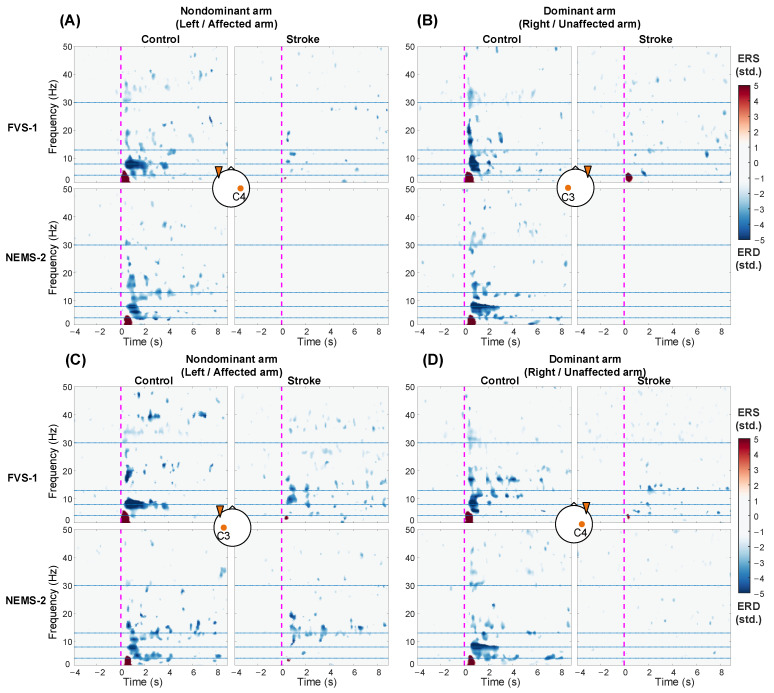
ERSP averaged across subjects in the C3/C4 channel contralateral (**A**,**B**) and ipsilateral (**C**,**D**) to the stimulated arms during representative FVS and NMES. The blue and red color schemes denote the ERD and ERS, respectively. Significant ERD/ERS in comparison with the resting baseline (*p* < 0.05, baseline permutation statistical method with false discovery rate correction). The vertical dashed magenta lines at 0 s mean the onset of stimulations. The horizontal dotted blue lines show the frequency band boundaries at 4, 8, 13, and 30 Hz. The brain icon at the center of each subfigure shows the stimulated nondominant/dominant arm by the orange triangle on one side and the observed contralateral or ipsilateral channel by the C3/C4 label.

**Figure 8 bioengineering-11-00286-f008:**
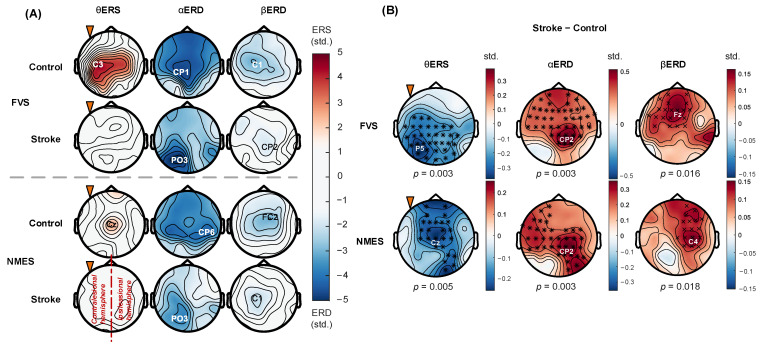
(**A**) ERD/ERS topographies when stimulating the nondominant arm. The blue and red color schemes denote the ERD and ERS, respectively. Peak channels are indicated by the labels. (**B**) Differences in topography between stroke and control groups. The blue and red color schemes denote the negative or positive differences, respectively. Significant differences between stroke and control groups are indicated by “×” for *p* < 0.05 and “*” for *p* < 0.01 (cluster-based permutation test). The orange triangle indicates stimulating the nondominant (left/affected) arms. The right hemisphere is ipsilesional for the stoke group.

**Table 1 bioengineering-11-00286-t001:** Demographic data of the participants.

Group	No. of Participants	Stroke Type (Hemorrhage/ Ischemic)	Affected Arm (Left/Right)	Gender (Male/Female)	Age (Years, Mean ± SD)	Years after Stroke (Min/Max Years)
Stroke	15	9/6	10/5	8/7	53 ± 11	2/18
Control	15	−/−	−/−	9/6	67 ± 3	−/−

**Table 2 bioengineering-11-00286-t002:** Clinical scores of the stroke participants.

Clinical Assessment	FMA	MAS		Monofilament			
	Upper Extremity	Wrist	Finger	Affected Extensor	Affected Flexor	Unaffected Extensor	Unaffected Flexor
Score (mean ± SD)	31.1 ± 11.5	1.1 ± 0.9	1.9 ± 0.8	3.23 ± 0.67	3.24 ± 0.49	3 ± 0.45	3.26 ± 0.38

**Table 3 bioengineering-11-00286-t003:** Levels and abbreviation labels of the protocol.

Group	Target Arm	Target Muscle Union	Stimulation Scheme/Trial
Stroke group	Nondominant/affected arm (A)	Affected ECU-ED (AE)	FVS-1,2,3,4,5; NMES-1,2,3
		Affected FCR-FD (AF)	FVS-1,2,3,4,5; NMES-1,2,3
	Dominant/unaffected arm (U)	Unaffected ECU-ED (UE)	FVS-1,2,3,4,5; NMES-1,2,3
		Unaffected FCR-FD (UF)	FVS-1,2,3,4,5; NMES-1,2,3
Control group	Nondominant/left arm (L)	Left ECU-ED (LE)	FVS-1,2,3,4,5; NMES-1,2,3
		Left FCR-FD (LF)	FVS-1,2,3,4,5; NMES-1,2,3
	Dominant/right arm (R)	Right ECU-ED (RE)	FVS-1,2,3,4,5; NMES-1,2,3
		Right FCR-FD (RF)	FVS-1,2,3,4,5; NMES-1,2,3

## Data Availability

All data from this study are available from the authors upon reasonable request.

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
