# Peer review of "Comparison of Immediate Neuromodulatory Effects between Focal Vibratory and Electrical Sensory Stimulations after Stroke"

_bioengineering, 2024, doi:10.3390/bioengineering11030286_

Round 1

Reviewer 1 Report

Comments and Suggestions for Authors

The paper reports of the comparison between cortical responses in chronic stroke patients and healthy controls measured by electroencephalography (EEG) in response to Focal vibratory stimulation (FVS) and neuromuscular electrical stimulation (NMES). The paper is well written, the methods are well described, and the topic is of great interest. In my opinion only few concerns need to be addressed before publication:

1)        The EEG filtering process is reported twice (lines 180 and 193), maybe this redundancy could be removed.

2)        In figure 3 the two section are separated according to the employment of parametric or not parametric statistical tests, however the t-test is mentioned also in the section associated with the not parametric tests. Please check.

3)        Please better describe in the results section where Bonferroni correction was used, specifying whether adjusted p-values are reported.

4)        Concerning the electrodes placement, reporting a figure with the schematic placement of all the electrodes could be beneficial for the readers.

5)        In the Discussion section it could be interesting to discuss, as future investigations, the possibility to assess brain activities associated with the use of distributed stimulations as the Whole Body Vibration and Transcutaneous Electrical Nerve Stimulation. Please refer to:

·        Perpetuini, D., Russo, E. F., Cardone, D., Palmieri, R., De Giacomo, A., Pellegrino, R., ... & Filoni, S. (2023). Use and Effectiveness of Electrosuit in Neurological Disorders: A Systematic Review with Clinical Implications. Bioengineering, 10(6), 680.

·        Celletti, C., Suppa, A., Bianchini, E., Lakin, S., Toscano, M., La Torre, G., ... & Camerota, F. (2020). Promoting post-stroke recovery through focal or whole body vibration: criticisms and prospects from a narrative review. Neurological Sciences41, 11-24.

Author Response

Thanks for the comments and recommendations from the reviewers and the Editorial Board. Revisions in the manuscript were highlighted in red fonts with underlines in the manuscript. The corresponding response to each comment is as follows:

Comment 1:   The EEG filtering process is reported twice (lines 180 and 193), maybe this redundancy could be removed.

Response: The EEG filtering process in line 180 was applied for EEG real-time monitoring during the experiment, while the EEG filtering process in line 193 was applied for offline pre-processing of the raw EEG data.

Revision: The descriptions of the respective real-time and offline EEG processing have been revised for clarification in line 183 and 197.

Comment 2:  In figure 3 the two sections are separated according to the employment of parametric or not parametric statistical tests; however, the t-test is mentioned also in the section associated with the not parametric tests. Please check.

Response & Revision: The P300 time-series signals did not follow a normal distribution and were compared using nonparametric statistical tests, i.e., cluster-based permutation tests. On the other hand, the P300 peak’s amplitude and latency data exhibited a normal distribution and were compared using parametric tests, i.e., t-tests. The incorrect categorization of the statistical methods has been rectified in Figure 3.

Comment 3:  Please better describe in the results section where Bonferroni correction was used, specifying whether adjusted p-values are reported.

Response & Revision: The Bonferroni correction was applied in the one-way ANOVA with repeated measures to compare the RSP among the eight stimulation schemes. The usage of Bonferroni correction has been specified in the corresponding results on page 11, lines 386 and 389. The statistical analysis was conducted using SPSS 24.0 (2016), and the reported p-values were all adjusted when the Bonferroni correction was used.

Comment 4:  Concerning the electrodes placement, reporting a figure with the schematic placement of all the electrodes could be beneficial for the readers.

Response & Revision: The NMES electrode layout on the forearm and the EEG electrode layout according to the standard 10-10 system [31] have been added in Figure A1 in the Appendix section.

Comment 5:  In the Discussion section it could be interesting to discuss, as future investigations, the possibility to assess brain activities associated with the use of distributed stimulations as the Whole Body Vibration and Transcutaneous Electrical Nerve Stimulation. Please refer to:

  • Perpetuini, D., Russo, E. F., Cardone, D., Palmieri, R., De Giacomo, A., Pellegrino, R., ... & Filoni, S. (2023). Use and Effectiveness of Electrosuit in Neurological Disorders: A Systematic Review with Clinical Implications. Bioengineering, 10(6), 680.
  • Celletti, C., Suppa, A., Bianchini, E., Lakin, S., Toscano, M., La Torre, G., ... & Camerota, F. (2020). Promoting post-stroke recovery through focal or whole body vibration: criticisms and prospects from a narrative review. Neurological Sciences, 41, 11-24.

Revision: “In addition to focal stimulation on a single target muscle, the application of distributed electrical and vibratory stimulations to multiple muscles has also demonstrated rehabilitative effects in patients following neurological disorders, e.g., whole-body vibration (WBV) [72] and the Electrosuit [73]. Future studies will be conducted on developing selective stimulation techniques with distributed vibratory stimulation and investigating the related neuromodulatory effects on muscle groups (e.g., agonist and antagonist, proximal and distal, etc.) with larger sample sizes of participants.” has been added to section 4.5.

Reviewer 2 Report

Comments and Suggestions for Authors

The manuscript, "Comparison of Immediate Neuromodulatory Effects between Focal Vibratory and Electrical Sensory stimulations after Stroke," has been reviewed.

 This interesting study concerns FVS and NMES stimuli, and some differences were determined by using EEG signals for stroke rehabilitation in the future. Therefore, scientific findings are relevant to defining the score of this work.

Despite this, Just some remarks need to be addressed.

 1—The main constraint in this study was the number of cases (15 stroke and 15 controls). How can authors justify and assess this work with this little number? Also, a brief description of the control volunteer information (other mental disorders, etc.) must be included.

 2. A brief description of hardware for data acquisition board (sample rate, data acquisition frequency) should be done.

 3. 2.2 paragraph: What was the total acquisition period for FVS and NMES? 10 seconds? or 10 minutes? It is not clear. Otherwise, a table relating to the data acquisition protocol, referencing Figure 1D, could be added.

 4. Please add some graphs obtained from the spectral analysis to the appendix session to show the differences between FVS and NMES, respectively.  

Comments on the Quality of English Language

Comment:

Moderate editing of English language required

Author Response

Thanks for the comments and recommendations from the reviewers and the Editorial Board. Revisions in the manuscript were highlighted in red fonts with underlines in the manuscript. The corresponding response to each comment is as follows:

Comment 1:  The main constraint in this study was the number of cases (15 stroke and 15 controls). How can authors justify and assess this work with this little number? Also, a brief description of the control volunteer information (other mental disorders, etc.) must be included.

Response: The primary aim of the study was to investigate the different cortical responses to FVS and NMES with different intensities in persons with chronic stroke, in comparison to those in the unimpaired controls. The recruitment of participants continued until significant differences in the key EEG parameters, such as P300, RSP, etc., were observed between FVS and NMES among the groups. The participants in the control group had no history of neurological, psychiatric, cardiovascular, cognitive impairments and mental disorders before the experiment. The study finally recruited 15 participants in each group, and the statistical significances achieved in the study showed sufficient effect sizes to reach conclusions. In future studies, a larger sample will be achieved for the investigation on the modulatory effects of FVS and NMES with distributed stimulation on different muscles poststroke.

Revision: “One potential limitation of this study is the relatively small sample size. The recruitment of participants continued until significant differences in the key EEG parameters were observed between FVS and NMES among the groups. Fifteen participants were finally recruited in each group, and the statistical significances achieved in the study showed sufficient effect sizes to reach conclusions.” and “Future studies will be conducted on developing selective stimulation techniques with distributed vibratory stimulation and investigating the related neuromodulatory effects on muscle groups (e.g., agonist and antagonist, proximal and distal, etc.) with larger sample sizes of participants.” have been added to the section 4.5.

“The inclusion criteria of the control group were right-handed and no history of neurological, psychiatric, cardiovascular, cognitive, or mental impairments.” has been added to section 2.1 Participants.

Comment 2:  A brief description of hardware for data acquisition board (sample rate, data acquisition frequency) should be done.

Revision: “During a stimulation trial, the EEG signals were amplified with a gain of 10,000, digitized at an analog-to-digital sampling rate of 1000 Hz.” has been added to section 2.3, line 182.

Comment 3:  2.2 paragraph: What was the total acquisition period for FVS and NMES? 10 seconds? or 10 minutes? It is not clear. Otherwise, a table relating to the data acquisition protocol, referencing Figure 1D, could be added.

Revision: “The acquisition duration was 4 minutes for each trial including the baseline and stimulation periods.” has been added to section 2.3, line 186.

Comment 4:  Please add some graphs obtained from the spectral analysis to the appendix session to show the differences between FVS and NMES, respectively. 

Revision: An additional figure regarding the comparison of RSP on different muscle unions has been added to Figure A2 in the Appendix section.

Reviewer 3 Report

Comments and Suggestions for Authors

Comparative study on changes in the sensorimotor cortex observed after a physiotherapy intervention based on Focal vibratory stimulation and NMES. A novel study, which addresses the impact of transcutaneous procedures, however, the authors do not show why these two procedures should be investigated before others. A review of the scientific literature is suggested.

The manuscript is well written, with well defined objectives and hypotheses, although the results should be taken with caution due to the small sample size. We recommend the authors to describe a section on sample size calculation.

On the other hand, we observe a significant difference between the mean age of the two population groups. This could also influence the results obtained. We recommend the authors to address this issue.

In section 2.2 Experimental setup, the model of the vibrating device is indicated, however, the technical characteristics indicated in the manuscript are limited. Has this device been used in previous studies? Has it shown efficacy in similar studies? Idem for the NMES device. What type of electric current and type of impulse have you used? Please clarify.

Author Response

Thanks for the comments and recommendations from the reviewers and the Editorial Board. Revisions in the manuscript were highlighted in red fonts with underlines in the manuscript. The corresponding response to each comment is as follows:

Comment 1:   Comparative study on changes in the sensorimotor cortex observed after a physiotherapy intervention based on Focal vibratory stimulation and NMES. A novel study, which addresses the impact of transcutaneous procedures, however, the authors do not show why these two procedures should be investigated before others. A review of the scientific literature is suggested.

Revision: “Neuromuscular electrical stimulation (NMES) and focal vibratory stimulation (FVS) are the primary techniques used to deliver additional somatosensory stimulation to specific muscles transcutaneously for sensorimotor rehabilitation after stroke [2,4,5].” has been added to the Introduction section.

Comment 2:   The manuscript is well written, with well defined objectives and hypotheses, although the results should be taken with caution due to the small sample size. We recommend the authors to describe a section on sample size calculation.

Response: The primary aim of the study was to investigate the different cortical responses to FVS and NMES with different intensities in persons with chronic stroke, in comparison to those in the unimpaired controls. The recruitment of participants continued until significant differences in the key EEG parameters, such as P300, RSP, etc., were observed between FVS and NMES among the groups. The study finally recruited 15 participants in each group, and the statistical significances achieved in the study showed sufficient effect sizes to reach conclusions. In future studies, a larger sample will be achieved for the investigation on the modulatory effects of FVS and NMES with distributed stimulation on different muscles poststroke. The above information has been included in the discussion of limitations.

Revision: “One potential limitation of this study is the relatively small sample size. The recruitment of participants continued until significant differences in the key EEG parameters were observed between FVS and NMES among the groups. Fifteen participants were finally recruited in each group, and the statistical significances achieved in the study showed sufficient effect sizes to reach conclusions.” and “Future studies will be conducted on developing selective stimulation techniques with distributed vibratory stimulation and investigating the related neuromodulatory effects on muscle groups (e.g., agonist and antagonist, proximal and distal, etc.) with larger sample sizes of participants.” have been added to the section 4.5.

Comment 3:   we observe a significant difference between the mean age of the two population groups. This could also influence the results obtained. We recommend the authors to address this issue.

Revision: “Another limitation of this study was the age disparity between the stroke and control groups. Despite the control group having a higher mean age than the stroke group, the significant inter-group differences in the EEG patterns remain evident. The significant findings suggested that neurological impairments introduced by stroke were the dominant factors over aging [71] on the cortical responses to these sensory stimuli.” has been added to the section 4.5.

Comment 4:   In section 2.2 Experimental setup, the model of the vibrating device is indicated, however, the technical characteristics indicated in the manuscript are limited. Has this device been used in previous studies? Has it shown efficacy in similar studies? Idem for the NMES device. What type of electric current and type of impulse have you used? Please clarify.

Response & Revision: The description of the NMES device has been modified in lines 149-152 to indicate that it generates alternating current in square pulses with a frequency of 40 Hz (i.e., 40 pulses per second), an amplitude of 70 V, and an adjustable pulse width ranging from 0 to 300 μs, allowing for different levels of stimulation intensity. Our previous studies have demonstrated the efficacy of the well-established NMES control box [27, 33, 36] (Qian et al., 2017; Rong et al., 2017). The same series of vibration motors used for FVS in this study have been utilized in previous studies on poststroke rehabilitation, demonstrating significant improvements in sensory and motor functions [22, 39] (Seim et al., 2023). “We integrated a one-channel FVS into an NMES control box that we developed previously [27, 33, 36].” and tenchincal characteristics of the FVS motor have been added to line 142.